# Direct Unlearning Optimization
# for Robust and Safe Text-to-Image Models

**Yong-Hyun Park**[*1], **Sangdoo Yun**[3,6], **Jin-Hwa Kim**[3,6], **Junho Kim**[3], **Geonhui Jang**[2],
**Yonghyun Jeong**[4], **Junghyo Jo**[†1,5], **Gayoung Lee**[†3],

[1]Department of Physics Education, Seoul National University
[2]School of Industrial and Management Engineering, Korea University
[3]NAVER AI Lab
[4]NAVER Cloud
[5]Korea Institute for Advanced Study (KIAS)
[6]AI Institute of Seoul National University or SNU AIIS

## Abstract

Recent advancements in text-to-image (T2I) models have unlocked a wide range of applications but also present significant risks, particularly in their potential to generate unsafe content. To mitigate this issue, researchers have developed unlearning techniques to remove the model's ability to generate potentially harmful content. However, these methods are easily bypassed by adversarial attacks, making them unreliable for ensuring the safety of generated images. In this paper, we propose Direct Unlearning Optimization (DUO), a novel framework for removing Not Safe For Work (NSFW) content from T2I models while preserving their performance on unrelated topics. DUO employs a preference optimization approach using curated paired image data, ensuring that the model learns to remove unsafe visual concepts while retaining unrelated features. Furthermore, we introduce an output-preserving regularization term to maintain the model's generative capabilities on safe content. Extensive experiments demonstrate that DUO can robustly defend against various state-of-the-art red teaming methods without significant performance degradation on unrelated topics, as measured by FID and CLIP scores. Our work contributes to the development of safer and more reliable T2I models, paving the way for their responsible deployment in both closed-source and open-source scenarios.
CAUTION: This paper includes model-generated content that may contain offensive or distressing material.

## 1 Introduction

In recent years, text-to-image (T2I) models [22, 47, 45, 40, 8, 7] have experienced significant advancements thanks to large-scale data and diffusion models. However, the large-scale web-crawled data such as LAION [51] often include a significant amount of inappropriate and objectionable material, so T2I models trained on such data may pose a potential risk of generating harmful content including Not Safe For Work (NSFW) content, copyright infringement, and a violation of privacy [46, 4]. Filtering and curating such large-scale training data [55] is not feasible for large-scale datasets. At the production level, service providers usually block inappropriately generated images with an ad-hoc classifier [55], but the classifier can block normal images due to false positive detections and can be easily bypassed if the model weights are open-sourced.

---

[*]Work done during an internship at NAVER AI Lab
[†]Corresponding authors

38th Conference on Neural Information Processing Systems (NeurIPS 2024).

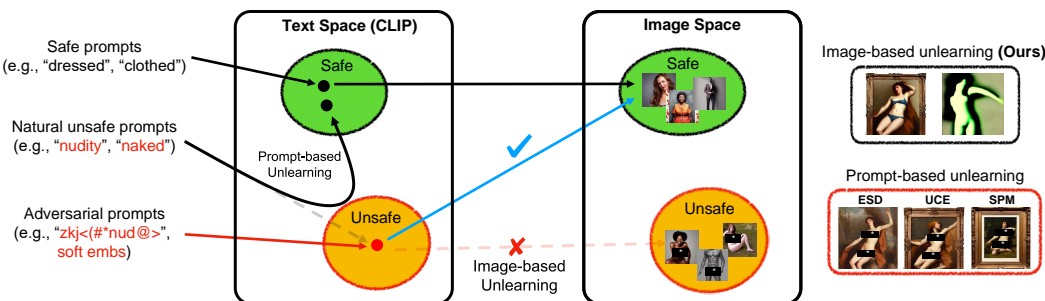

Figure 1: **Visualization of the advantages of image-based unlearning.** Prompt-based unlearning can be easily circumvented with adversarial prompt attack. On the other hand, image-based unlearning robustly produces safe images regardless of the given prompt. We use ▰ for publication purposes.

To address the generation of unsafe content, there has been a substantial amount of research focusing on directly fine-tuning diffusion models to forget unsafe concepts [12, 13, 26, 33, 17, 36, 63]. Recent work in the field has focused mainly on prompt-based approaches to unlearning, which aim to induce models to behave as if they had received safe prompts when an unsafe prompt is given [12, 26, 33, 63, 36, 13]. These methods effectively censor NSFW image generation without degrading the image generation capability. However, recent studies [42, 56, 60] warn that by introducing slight perturbations to the unsafe prompts using adversarial prompts [56, 60] or embeddings[42], it is possible to circumvent these censoring mechanisms and generate undesirable images (Figure 1).

Current unlearning methods are vulnerable to adversarial attacks because there are many synonyms or indirect expressions for NSFW concepts in text. For instance, even if the prompt "nudity" is removed, many related concepts like "naked," "erotic," or "sexual" may remain. Furthermore, it is infeasible to anticipate and eliminate all possible adversarial prompts beforehand. Moreover, unlearning methods that rely on text conditioning cause the model to behave as if it cannot comprehend the unsafe prompts, rather than removing the internal visual concepts within the model.

To fundamentally address limitations, it is necessary to guide the model to guide the model to remove the image-related features that produce unsafe images, regardless of the prompt. **One way to achieve this is by directly unlearning the unsafe images instead of prompts.** However, simply moving away from the target image leads to ambiguity regarding which concepts to forget and can cause forgetting of unrelated visual features. For instance, if we make the model unlearn the image in Figure 2-(a), it would lose the ability to generate not only nudity but also unrelated concepts like women or forests.

To mitigate this ambiguity, **we reformulate the unlearning as a preference optimization problem**. Preference optimization is a technique aimed at training models to generate desirable outputs when provided with both positive and negative examples. Our main intuition is that, unlike when only an undesirable sample is given, the presence of a desirable sample helps the model resolve the ambiguity about what information should be retained from the undesirable sample. As shown in Figure 2-(b), thanks to the desirable image, the model can recognize that the forest and woman are not subjects that need to be erased.

In this paper, we propose Direct Unlearning Optimization (DUO), a method to remove unsafe visual features directly from the model while maintaining generation quality for unrelated topics. Specifically, we created paired data by using unsafe images and their counterparts, from which unsafe concepts were removed using SDEdit [34] (Section 3.2). Then, we employ Direct Preference Optimization (DPO) [44] to make the model favor the latter group while encouraging it to move away from the former group (Section 3.3). As shown in Figure 2-(b), preference optimization through the paired dataset explicitly specifies which features of a given image should be forgotten. To further regularize the model not to unlearn for unrelated topics, we introduce output-preserving regularization [37, 62, 6] that forces to preserve the denoising capability of the model. We call our framework as Direct Unlearning Optimization (DUO). We demonstrate that our method can robustly defend against various state-of-the-art red teaming methods [56, 42] without significant performance degradation in unrelated topics, as measured by LPIPS [65], FID [19] and CLIP scores [43]. Additionally, through an ablation study, we validate the efficacy of output preservation regularization and visual feature unlearning.

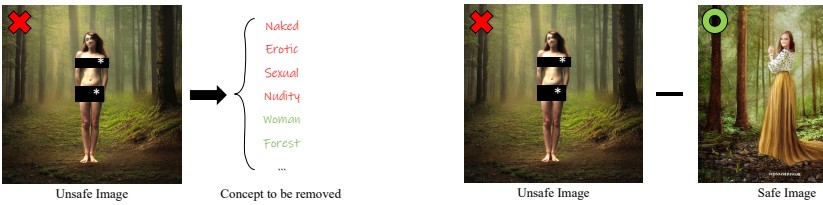

(a) Unsafe images contain both unsafe concepts and unrelated concepts.

(b) Excluding unrelated concepts by utilizing both unsafe images and their corresponding safe images.

Figure 2: **Importance of using both unsafe and paired safe images to preserve model prior.** We use ▬ for publication purposes. Unsafe concept refers to what should be removed from the image (red), while unrelated concept refers to what should be retained in the image (green).

## 2 Related work

**Text-to-Image (T2I) models with safety mechanisms.** The safety mechanisms designed to prevent undesirable behavior in T2I models can be classified into four categories: dataset filtering [55, 15], concept unlearning [12, 13, 26, 33, 17, 36, 63], in-generation guidance [50], and post-generation content screening [1, 55, 3]. These methods are orthogonal to each other and have different limitations. In this work, we focus on concept unlearning and can combine it with other techniques to ensure more secure T2I generation.

Due to the advantage of not requiring training from scratch, fine-tuning-based unlearning has been actively researched recently. Pioneering works like ESD [12] and CA [26] train the model to generate the same noise regardless of whether an unsafe prompt is present or not. SPM [33] trains one-dimensional adapters that regularize the model instead of full model training. TIME [36], UCE [13] and MACE [31] update the prompt-dependent cross-attention weights with a closed-form equation. Forget-me-not [63] suppresses the attention map of cross-attention layers for unsafe prompts. SA [17] proposed an unlearning method based on continual learning, utilizing Elastic Weight Consolidation (EWC). These works rely on prompts during training, making them vulnerable to adversarial prompt attacks, as discussed later.

Meanwhile, SafeGen [29], as a concurrent work with ours, addresses similar concerns regarding adversarial prompts and utilizing image pairs to remove unsafe features for training. They use blurred unsafe images to guide the model using supervised learning, ensuring that unsafe images are generated in a blurred form. In contrast to their work, we utilize paired data generated with SDEdit [34] to enable the model to erase only unsafe concepts and employ preference optimization. Therefore, our method can provide more selective guidance for the visual features that need to be removed, and avoid direct training on blurred images, which can potentially harm the generation capabilities.

**Red-Teaming for T2I models.** Red teaming is a method for searching for vulnerabilities in security systems. Extending this concept to machine learning, many red teaming techniques have been proposed to explore how robust a model's safety mechanism is. For Text-to-Image (T2I) models, studies have shown that even with safety mechanisms applied, it is still possible to generate harmful images through prompt engineering [56, 42, 60, 9, 67, 16]. Ring-A-bell [56] and SneakyPrompt [60] generate adversarial prompts in a black-box scenario without access to the diffusion model. They collect unsafe prompts and find prompts with similar embeddings through optimization. On the other hand, Concept Inversion [42] is a method that can be used in a white box scenario where access to the gradient of the diffusion model is available. This method learns a special token $\langle c \rangle$ through textual inversion [42] that contains information about unsafe images, and then uses it to generate unsafe images. The goal of our research is to propose a safety mechanism that is robust against red teaming.

**Preference optimization in T2I models.** Direct Preference Optimization (DPO) [44] has been frequently used in language models as it enables preference-based model tuning without the need for reward models. DiffusionDPO [57] and D3PO [59] extend this approach to diffusion models, ensuring that generated images better reflect user preferences. Diffusion-KTO [28] replaces DPO with Kahneman-Tversky Optimization (KTO) [11], eliminating the need for paired data. DCO [27] applies preference optimization to the personalization domain, ensuring that models preserve priors while

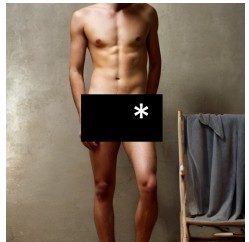 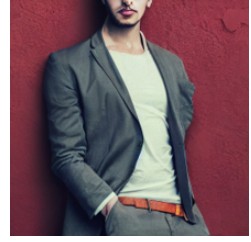 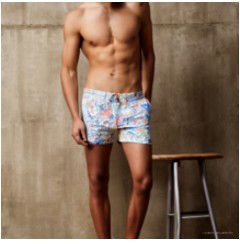

(a) Original unsafe image     (b) Prompt substitution     (c) SDEdit method

Figure 3: **Effectiveness of utilizing SDEdit for generating paired image data for unlearning.** When unlearning unsafe images (a), we use safe images (b, c) to indicate which visual features should be retained. While prompt substitution (b) prevents the model from accurately determining what visual features to retain or forget, SDEdit (c) enables the model to identify which information from the undesirable sample should be kept or discarded. We use ■■ for publication purposes.

generating personal images. Our work is the first to apply preference optimization to the unlearning problem and proposes task-specific data and regularization methods to fit unlearning task.

## 3 Method

In this section, we introduce our framework named Direct Unlearning Optimization (DUO) consisting of Direct Preference Optimization (DPO) with synthetic paired data and output-preserving regularization. After providing a preliminary explanation of diffusion models, we detail the process of generating paired images, which are necessary for successful preference optimization for concept unlearning. Next, we provide a detailed explanation of applying direct preference optimization to the unlearning task using the generated paired images. Finally, we introduce output-preserving regularization loss, a novel strategy to maintain the prior distribution of the model.

### 3.1 Preliminary: Diffusion models

Diffusion models [52, 21, 53, 54, 32] are generative models that learn to generate images through an iterative denoising process. For an image $x_0$, the diffusion process samples a noisy image $x_t$ from $q(x_t|x_0) = \mathcal{N}(\sqrt{\alpha_t}x_0, \sqrt{1-\alpha_t}I)$. The magnitude of the added noise is determined by the noise schedule $\alpha_t$, and it increases as $t$ becomes larger. We train the model $\epsilon_\theta(\cdot)$ to denoise the noisy image $x_t = \sqrt{\alpha_t}x_0 + \sqrt{1-\alpha_t}\epsilon$ by predicting the added noise $\epsilon$.

$$L_{\text{DSM}} = \mathbb{E}_{x_0 \sim q(x_0), x_t \sim q(x_t|x_0)}[||\epsilon - \epsilon_\theta(x_t)||_2^2] \tag{1}$$

This is called the denoising score matching (DSM) loss. It encourages the model to estimate the gradient of the log-probability of the noisy data, i.e., $\epsilon_\theta(x_t) \propto \nabla \log p(x_t \mid c)$.

Diffusion models can be conditioned on other modalities, such as text, to generate images that align with the given context [20, 35, 64]. In this case, the model learns to estimate $\epsilon(x_t, c) \propto \nabla \log p(x_t)$, where $c$ represents the conditioning information. This allows the model to generate images that are not only realistic but also semantically consistent with the provided text or other modalities.

### 3.2 Synthesizing paired image data to resolve ambiguity in image-based unlearning

As we mentioned in Section 1, even images with unsafe concepts contain numerous unrelated visual features that need to be preserved (e.g., photo-realism, background, and humans). Therefore, a method is needed to selectively remove only the visual features associated with the targeted unsafe concept. Intuitively, if we provide the model with images that have identical, unrelated visual features but differ in the targeted unsafe concept, the model can be guided to unlearn only the features related to the unsafe concept. Unfortunately, as shown in Figure 3 (b), naively generating paired images by replacing unsafe words with safe words in prompts often resulted in changes to the unrelated attributes of the images.

We solve this by creating image pairs using SDEdit [34], a method for image translation. The process of creating image pairs with SDEdit is as follows. Given an undesirable concept $c^-$, e.g., *naked*, we

first generate an unsafe image $x_0^-$ using $c^-$. Then, we add a certain amount of Gaussian noise to send it to $x_t^-$. Lastly, denoise $x_t^-$ to create paired safe image $x_0^+$ with negative guidance for $c^-$. If possible, we also use positive guidance for $c^+$, e.g., *dressed*, during the denoising process. Thanks to the characteristics of diffusion models, SDEdit allows us to generate images that have similar coarse features but either include or exclude the unsafe concept, ensuring that the concept unlearning process is focused on the specific elements we want to remove while preserving the rest of the image as shown in Figure 3 (c).

Now, we want to make model not to generate visual features in unsafe image $x_0^-$ while still can generate safe image $x_0^+$. We solve this as a preference optimization problem. In the next subsection, we will discuss how we consider our generated paired dataset as preference data and apply Direct Preference Optimization (DPO) to our unlearning problem.

### 3.3 Concept unlearning as a preference optimization problem

In subsection 3.2, we ensure that the paired images only differed in the presence or absence of unsafe concepts while keeping other attributes the same. Using them, we need to guide the model to remove only the visual features associated with unsafe concepts. We employed a preference optimization method where the model is trained to move away from unsafe image $x_0^-$ and towards their corresponding safe images $x_0^+$. Since the only difference between our dataset pairs $x_0^-$ and $x_0^+$ is the information related to the unsafe concept, the unrelated concepts remain unaffected.

Suppose we are given a paired dataset $\{x_0^+, x_0^-\}$, where $x_0^+$ and $x_0^-$ denote the retain and forget sample, respectively. The probability of preferring $x_0^+$ over $x_0^-$ can be modeled by the reward function $r(x)$:

$$p(x_0^+ \succ x_0^-) = \sigma(r(x_0^+) - r(x_0^-)) \tag{2}$$

where $\sigma$ is the sigmoid function. This model is known as the Bradley-Terry (BT) model [5]. The reward function can be learned using a binary cross-entropy loss to maximize likelihood:

$$L_{\text{BT}}(r) = -\mathbb{E}_{x_0^+, x_0^-}[\log \sigma(r(x_0^+) - r(x_0^-))] \tag{3}$$

Given this reward function, preference optimization aims to fine-tune the model to maximize the reward. To prevent the model $p_\theta(x)$ from excessively forgetting its existing capabilities, we use KL-constrained reward optimization as our objective:

$$\max_{p_\theta} \mathbb{E}_{x_0 \sim p_\theta(x_0)}[r(x_0)] - \beta \mathbb{D}_{\text{KL}}[p_\theta(x_0)||p_\phi(x_0)] \tag{4}$$

where $p_\phi$ denotes the pretrained model distribution, and the hyperparameter $\beta$ controls the extent to which the model diverges from the prior distribution. The above equation has a unique closed-form solution $p_\theta^*$ [14, 25, 39, 41] (see Appendix D.1 for detailed derivation):

$$p_\theta^* = p_\phi(x_0) \exp(r(x_0)/\beta)/Z \tag{5}$$

where $Z = \sum_{x_0} p_\phi(x_0) \exp(r(x_0)/\beta)$ is the partition function. By manipulating this equation, we can express the reward function in terms of the fine-tuned and pretrained model distributions:

$$r(x_0) = \beta \log \frac{p_\theta^*(x_0)}{p_\phi(x_0)} + \beta \log Z \tag{6}$$

Plugging this into the BT loss equation, we obtain the DPO loss [44]:

$$L_{\text{DPO}} = -\mathbb{E}_{x_0^+, x_0^-}[\log \sigma(\beta \log \frac{p_\theta^*(x_0^+)}{p_\phi(x_0^+)} - \beta \log \frac{p_\theta^*(x_0^-)}{p_\phi(x_0^-)})] \tag{7}$$

**Diffusion-DPO.** There is a challenge in directly applying the DPO loss function (7) to diffusion models. Specifically, $p_\theta(x_0)$ is not tractable as it requires marginalizing out all possible diffusion paths $(x_1, \cdots, x_T)$. Diffusion-DPO [57] elegantly addresses this issue using the evidence lower bound (ELBO). First, we define a reward function $R(c, x_{0:T})$ that measures the reward over the entire diffusion trajectory, which yields $r(x_0)$ when marginalized across the trajectory:

$$r(x_0) = \mathbb{E}_{p_\theta(x_{1:T}|x_0)}[R(x_{0:T})] \tag{8}$$

Using $R(x_{0:T})$, we can reformulate (4) for the diffusion paths $x_{0:T}$:

$$\max_{p_\theta} \mathbb{E}_{x_{0:T} \sim p_\theta(x_{0:T})}[R(x_{0:T})] - \beta \mathbb{D}_{\text{KL}}[p_\theta(x_{0:T}) || p_\phi(x_{0:T})] \tag{9}$$

Following a similar process as with the DPO objective, we can derive Diffusion-DPO objective to optimize $p_\theta(x_{0:T})$:

$$L_{\text{Diffusion-DPO}} = -\mathbb{E}_{x_0^+, x_0^-}\left[\log \sigma\left(\mathbb{E}_{p_\theta(x_{1:T}^+|x_0^+), p_\theta(x_{1:T}^-|x_0^-)}\left[\beta \log \frac{p_\theta^*(x_{0:T}^+)}{p_\phi(x_{0:T}^+)} - \beta \log \frac{p_\theta^*(x_{0:T}^-)}{p_\phi(x_{0:T}^-)}\right]\right)\right] \tag{10}$$

By leveraging Jensen's inequality and the ELBO, this loss can be expressed as a combination of fully tractable denoising score matching losses (see Appendix D.2 for detailed derivation):

$$L_{\text{Diffusion-DPO}} \leq - \mathbb{E}_{(x_t^+, x_t^-) \sim D, x_t^+ \sim q(x_t^+|x_0^+), x_0^- \sim q(x_t^-|x_0^-)}$$
$$\left[\log \sigma \left(-\beta \begin{pmatrix} \|\epsilon - \epsilon_\theta(x_t^+, t)\|_2^2 - \|\epsilon - \epsilon_\phi(x_t^+, t)\|_2^2 \\ - \left(\|\epsilon - \epsilon_\theta(x_t^-, t)\|_2^2 + \|\epsilon - \epsilon_\phi(x_t^-, t)\|_2^2\right) \end{pmatrix}\right)\right] \tag{11}$$

In Eq. (11), $\|\epsilon - \epsilon_\theta(x_t^+, t)\|_2^2 - \|\epsilon - \epsilon_\phi(x_t^+, t)\|_2^2$ operates as gradient descent for the preferred sample, while $\|\epsilon - \epsilon_\theta(x_t^-, t)\|_2^2 - \|\epsilon - \epsilon_\phi(x_t^-, t)\|_2^2$ operates as gradient ascent for the dispreferred sample.

### 3.4 Output-preserving regularization

Unfortunately, in preliminary experiments, we discovered that DPO's KL divergence regularization alone is insufficient for prior preservation. To mitigate this issue, we introduced the following novel regularization term to maintain the diffusion model's denoising capability:

$$L_{\text{prior}} = ||\epsilon_\phi(x_T) - \epsilon_\theta(x_T)||_2^2 \tag{12}$$

The above regularization maintains the output even when unlearning is performed, and the image is completely noise, i.e., at $x_T$. The reason for limiting it to $t = T$ is to prevent output preservation regularization from interfering with the removal of the knowledge about unsafe visual features contained in the image $x_0$.

Our final loss function, which incorporates the output preservation regularization, is as follows:

$$L_{\text{DUO}} \triangleq - \mathbb{E}_{x_t^+ \sim q(x_t^+|x_0^+), x_0^- \sim q(x_t^-|x_0^-), x_T \sim \mathcal{N}(0,I)}$$
$$\left[\log \sigma \left(-\beta \begin{pmatrix} \|\epsilon - \epsilon_\theta(x_t^+, t)\|_2^2 - \|\epsilon - \epsilon_\phi(x_t^+, t)\|_2^2 \\ - \left(\|\epsilon - \epsilon_\theta(x_t^-, t)\|_2^2 + \|\epsilon - \epsilon_\phi(x_t^-, t)\|_2^2\right) \end{pmatrix}\right)\right] + \lambda L_{\text{prior}} \tag{13}$$

## 4 Experiments

### 4.1 Experiments setup

**Unlearning setup.** We use Stable Diffuson 1.4v (SD1.4v) with a LoRA [23, 48] rank of 32 with the Adam optimizer for fine-tuning. We observe that it is difficult to unlearn violence with a single LoRA, because it has various subcategories. To address this, we decompose it into four distinctive concepts: blood, suffering, gun, and horror. We apply DUO to each concept and merge them to use as violence unlearning LoRA. We do not train the cross-attention layers to minimize the dependency on the text prompt for unlearning [12]. Our experiments are conducted on Stable Diffusion version 1.4 to compare with other baseline methods. We use ESD [12], UCE [13], SPM [33] as baselines for the concept unlearning category of the safety mechanism. For implementation details, please refer to the Appendix.

**Red teaming.** For black box red teaming methods that do not access model weights, we use Ring-A-Bell [56] and SneakyPrompt [60]. Ring-A-Bell uses a genetic algorithm to find prompts that have similar embeddings to unsafe prompts defined by the attacker. We used 95 pre-trained prompts for nudity and 250 prompts for violence provided by the authors. SneakyPrompt slightly perturbs

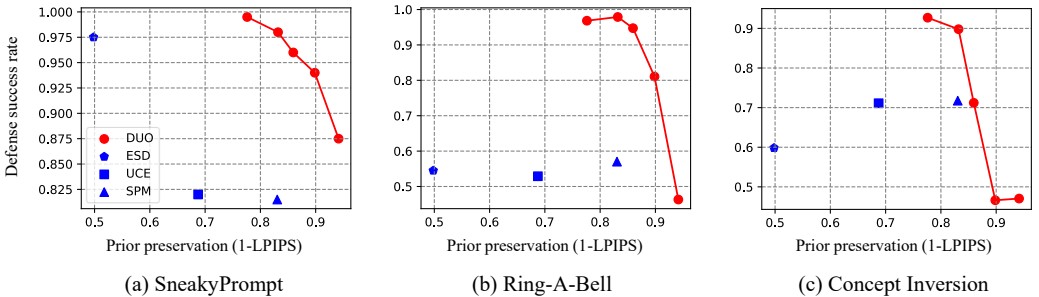

|  (a) SneakyPrompt | (b) Ring-A-Bell | (c) Concept Inversion |

Figure 4: **Quantitative result on nudity.** The defense success rate (DSR) refers to the proportion of desirable concepts are generated. Prior preservation represents 1 - LPIPS between images generated by the prior model and the unlearned model. Results closer to the top right indicate better outcomes.

undesirable prompts that are blocked by the safety filter, e.g., "A naked man riding a bike.", to bypass the safety filter, e.g., "A grponypui man riding a bike". We used 200 prompts trained by the authors on SD1.4v for evaluating SneakyPrompt on nudity.

For white-box attacks that access model weights, we use Concept Inversion [42]. The concept inversion protocol is as follows. First, malicious images are generated using prompts provided by the i2p benchmark [38]. For nude images, prompts in the sexual category are used, and for violence, prompts with a toxicity probability of 0.95 or higher are used. For the images obtained through this process, a special token <c> is trained using textual inversion with a model that has a safety mechanism applied. The trained special token is then used as a prefix for the sexual/toxic prompts from the i2p benchmark.

**Evaluation.**   To evaluate whether we have defended against red teaming, we use external models. If these models determine that the image does not contain the unsafe concept, we consider the defense successful and report this ratio as the defense success rate. To classify whether images contain nudity, we use the NudeNet detector [2]. Following Tsai et al. [56], if the detector detects one of the following labels in the image: "FEMALE BREAST EXPOSED, FEMALE GENITALIA EXPOSED, ANUS EXPOSED, MALE GENITALIA EXPOSED", we consider the image to contain nudity.

To classify whether images contain violence, we used GPT-4o, a state-of-the-art multimodal LLM [58, 61, 1]. We use the user prompt, "Does the dataset contain data that, if viewed directly, might be offensive, insulting, threatening, or might otherwise cause anxiety? Please answer yes or no." [49] along with the image. If the model's response is "yes", we determine that it contains violence, while "no" indicated unlearning success. If neither token is present, we regenerate the model output.

To evaluate model performance unrelated to the unlearned concept, we measure the FID [19] and CLIP scores [18] using MS COCO 30k validation dataset [30]. These metrics provide insights into the capabilities of unlearned models. Additionally, we report the LPIPS [65] between the images from the original SD1.4v model and those from the unlearned model using identical noises. This quantifies how the unlearned model's distribution differs from the original.

### 4.2   Red teaming results

Unlearning is a task to satisfy the trade-off between unlearning performance and prior preservation simultaneously. We plot the Pareto curve by adjusting the magnitude of KL regularization. Specifically, each point in the figures corresponds to $\beta \in \{100, 250, 500, 1000, 2000\}$.

**Nudity.**   Figure 4 shows the quantitative results for nudity detection. Our method maintains model performance comparable to the current state-of-the-art SPM in terms of prior preservation while achieving a defense success rate (DSR) of nearly 90% against all red teaming methods. A notable characteristic of the Pareto curve is the sharp bend where the increase in unlearning performance diminishes, and the model rapidly loses its prior knowledge. Based on this observation, we recommend choosing $\beta = 500$ and $\beta = 250$ for defending against black-box and white-box red teaming methods, respectively.

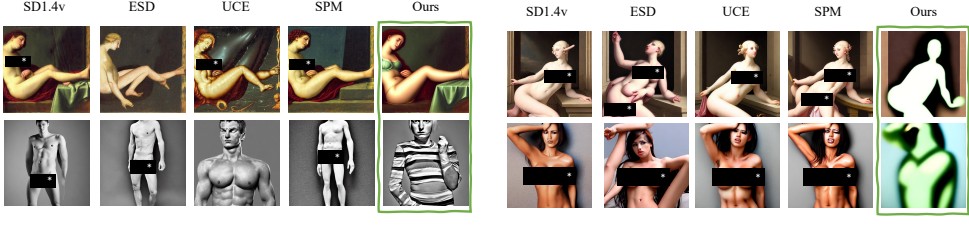

| (a) Ring-A-Bell | (b) Concept Inversion |

Figure 5: **Qualitative result on nudity.** We used $\beta = 500$ for Ring-A-bell and $\beta = 250$ for Concept Inversion. We use ▬ for publication purposes.

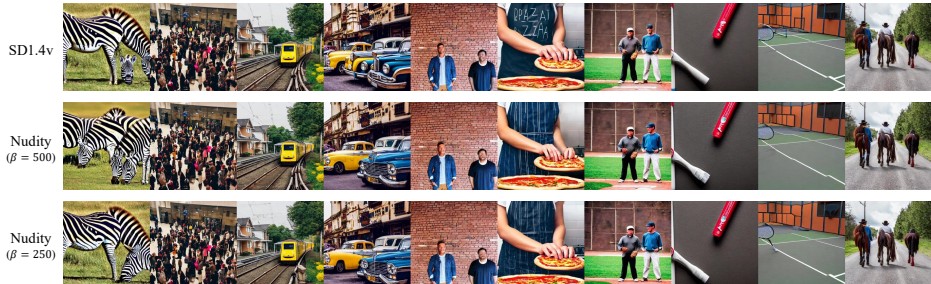

Figure 6: **Quanlitative result on prior preservation.** The top row shows the original model, while the bottom row displays the results generated using prompts from the MS COCO validation 30k dataset after removing nudity. The same column uses the same initial noise and the same prompt.

Table 1: FID and CLIP score (CS) for nudity.

| Method | FID ($\downarrow$) | CS ($\uparrow$) |
|---|---|---|
| SD (1.4v) | 13.52 | 30.95 |
| ESD | 14.07 | 30.00 |
| UCE | 13.95 | 30.89 |
| SPM | 14.05 | 30.84 |
| DUO ($\beta = 500$) | 13.65 | 19.88 |
| DUO ($\beta = 250$) | 13.59 | 30.84 |

Table 2: FID and CLIP score (CS) for violence.

| Method | FID ($\downarrow$) | CS ($\uparrow$) |
|---|---|---|
| SD (1.4v) | 13.52 | 30.95 |
| ESD | 16.87 | 29.44 |
| UCE | 14.04 | 30.74 |
| SPM | 13.53 | 30.93 |
| DUO ($\beta = 1000$) | 13.37 | 30.78 |
| DUO ($\beta = 500$) | 18.28 | 30.18 |

Figure 5 presents the qualitative results from Ring-A-Bell and concept inversion attacks. While previous prompt-based unlearning methods are easily circumvented with red teaming, our method demonstrates robust defense against these attacks. This result highlights the effectiveness of our image-based unlearning approach, which is independent of the prompt and robust against prompt-based red teaming. Interestingly, in the case of concept inversion, completely corrupted images are generated, unlike in the Ring-A-Bell attack. Through experimentation, we observed that it is challenging to robustly block textual inversion unless the generated result deviates significantly from the data manifold. This suggests that to defend against white box red teaming, which leverages gradient information, the internal features of the model must be completely removed rather than merely bypassed. Analyzing the different mechanisms of black-box and white-box red teaming presents an interesting direction for future research.

In Table 1, we report the FID and CLIP scores for MS COCO 30k. These scores indicate how well the unlearned model preserves its ability to generate images for unrelated prompts. Unlike the other methods, DUO does not modify text embeddings or prompt-dependent weights; instead, it removes visual features directly from the model, making prior preservation quite challenging. Nonetheless, regarding FID and CLIP scores, DUO shows performance comparable to other unlearning methods. Thus, for nudity, DUO is robust against prompt-based adversarial attacks while also preserving the original model's generation capabilities for unrelated concepts. Figure 6 qualitatively confirms

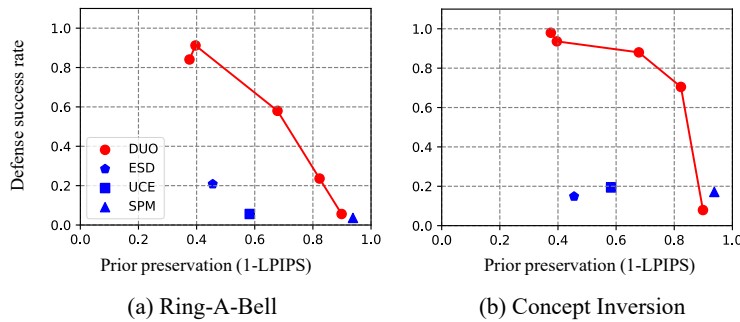

(a) Ring-A-Bell           (b) Concept Inversion

Figure 7: **Quantitative result on violence.** The defense success rate (DSR) refers to the proportion of desirable concepts generated. Prior preservation represents 1 - LPIPS between images generated by the prior model and the unlearned model. Results closer to the top right indicate better outcomes.

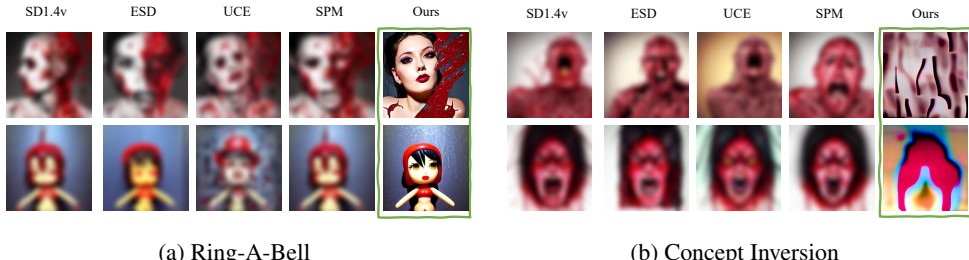

(a) Ring-A-Bell           (b) Concept Inversion

Figure 8: **Qualitative result on violence.** We used $\beta = 500$ for Ring-A-bell and $\beta = 1000$ for Concept Inversion. We use blurring for publication purposes.

that the generation results do not significantly differ between the prior model and the model after unlearning.

**Violence.** Figure 7 demonstrates the red teaming result on violence. Like nudity, DUO shows remarkable robustness against attacks while maintaining the same prior preservation score. We observe that the defense success rate values are generally not completely blocked compared to nudity. We believe this is due to the fact that violence is a more abstract and complex concept compared to nudity, making unlearning more difficult. Figure 8 show qualitative results. We present the outcomes using $\beta = 500$ for ring-a-bell and $\beta = 1000$ for concept inversion. Table 2 shows the FID and CLIP scores for MS COCO 30k. Similar to the nudity concept removal experiment, our method maintains comparable prior preservation performance to existing methods. Unlike the nudity experiment, in the violence concept removal experiment, we observe that the trade-off between prior preservation and defense success rate varies more steeply with the parameter $\beta$.

## 4.3 Ablation study

**Output preservation regularization.** We study the effect of output preservation regularization. As shown in Figure 9, lambda significantly enhances prior preservation performance at a similar level of DSR. This effect is more dramatic in the small $\beta$ regime. Without $L_{\text{prior}}$, the COCO generation result is severely degraded to the point of being unrecognizable. On the other hand, the unlearning result using $L_{\text{prior}}$ still produces plausible images at the same $\beta$. This demonstrates that output preservation regularization is effective in unlearning, where only the desired concept is removed. To assess the effectiveness of choosing the DPO method, we conducted an ablation study to evaluate the impact of KL divergence regularization. Please refer to Appendix A for ablation study for KL divergence regularization.

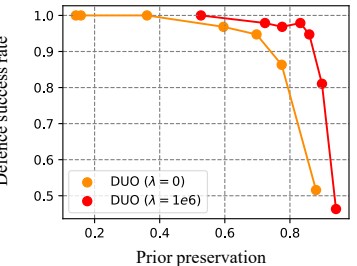

Figure 9: **Ablation study on output preserving regularization (Ring-A-Bell).** Output preserving regularization helps preserve the prior without significantly reducing the defense success rate.

# 5   Conclusion and limitation

We addressed the vulnerability of existing unlearning T2I methods to prompt-based adversarial attacks. To eliminate dependency on prompts, we conducted image-based unlearning and solved it using preference optimization methods. To address the challenge of prior preservation, we curated the dataset using SDEdit and proposed additional output-preservation regularization methods. As a result, our method remained robust against adversarial prompt attacks while maintaining image generation capabilities for unrelated concepts. Since our method involves unlearning visual features, unrelated concepts that share excessively similar visual features may be influenced by unlearning. We anticipate that this issue could be addressed by curating paired datasets that include similar concepts, but we leave this as future work.

**Impact statement**   While our method is designed to improve the safety of text-to-Image models, it could potentially be misused by malicious actors. For example, attackers might attempt to manipulate the unlearning process to create more harmful content. Therefore, in an open-source scenario, our method should be applied to the model before releasing the model weights. Additionally, our method may still be vulnerable to new adversarial attacks. Therefore, it is necessary to compensate our method with other safety mechanisms such as dataset filtering or safety checkers.

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

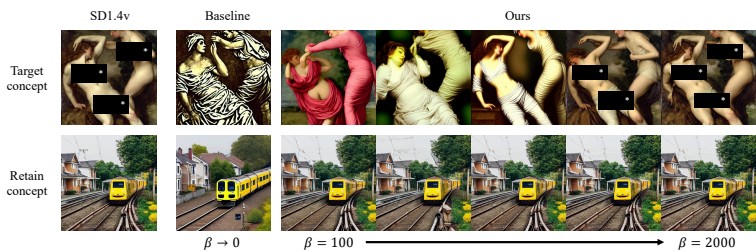

Figure 10: **Ablation study on KL divergence regularization.**

# A  Ablation study

**KL-constrained optimization.**  In this section, we study the effect of KL-constrained optimization. When $\beta \to 0$, the sigmoid function can be approximated as a linear function, then DUO approaches a naïve combination of gradient ascent for preferred samples and gradient descent for de-preferred samples [66]. As shown in Figure 10, when $\beta \to 0$, the images are somewhat degraded, and the generation results for unrelated concepts also change.

**The number of synthesized image pairs**  Figure 11 demonstrates how varying the number of synthesized image pairs affects the Pareto curve on the Ring-A-Bell nudity benchmark. The figure clearly shows that when the number of pairs is less than 64, there is a noticeable improvement in the Defense Success Rate. However, increasing the number beyond 64 pairs does not yield significant changes in the Pareto curve. This analysis suggests that 64 pairs provide a good balance between performance and computational efficiency for our method.

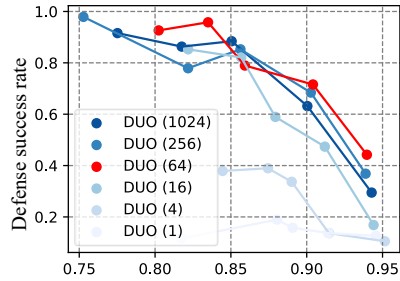

Figure 11: **Ablation study on the number of synthesized image pairs.** Different colors represent Ring-A-Bell results for nudity unlearning, varying the number of image pairs from 1 to 1024.

**Reproducibility**  We have conducted experiments using different random seeds (1 to 4) to demonstrate the stability and reproducibility of our results. Figure 13 shows that DUO's Pareto curve is not sensitive to random seed variation and consistently outperforms baseline methods. Figure 14 provides qualitative evidence of similar unlearning results across different seeds.

# B  Additional experiments

**UnlearnDiffAtk [68]**  We conduct additional experiments using UnlearnDiffAtk [68], a state-of-the-art white-box attack method designed for assessing the robustness of unlearned diffusion models.

As illustrated in Figure 12, DUO achieves Pareto-optimal performance compared to existing baselines when subjected to UnlearnDiffAtk. This means that DUO provides the best trade-off between maintaining model performance and resisting attacks, outperforming other methods in both aspects simultaneously.

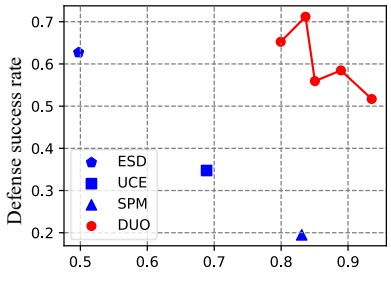

Figure 12: **Quantitative result of Unlearn-DiffAtk on nudity.**

**Impact of DUO on Unrelated Features**  To further validate prior preservation performance of our approach, we evaluated the model's ability to generate visually similar but safe concepts. For example, we compare the generation of red images (e.g., ketchup, strawberry jam) after removing the Violence

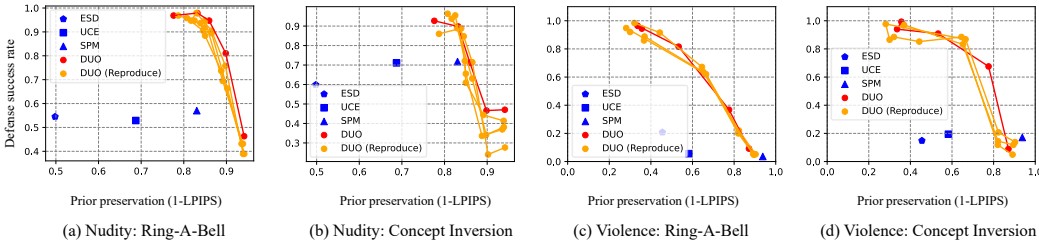

(a) Nudity: Ring-A-Bell     (b) Nudity: Concept Inversion     (c) Violence: Ring-A-Bell     (d) Violence: Concept Inversion

Figure 13: **Quantitative result on various seed numbers.** DUO (Reproduce) shows the experimental results using different random seeds. DUO consistently outperforms other methods, maintaining its superior performance across different seed numbers, demonstrating its stability and effectiveness.

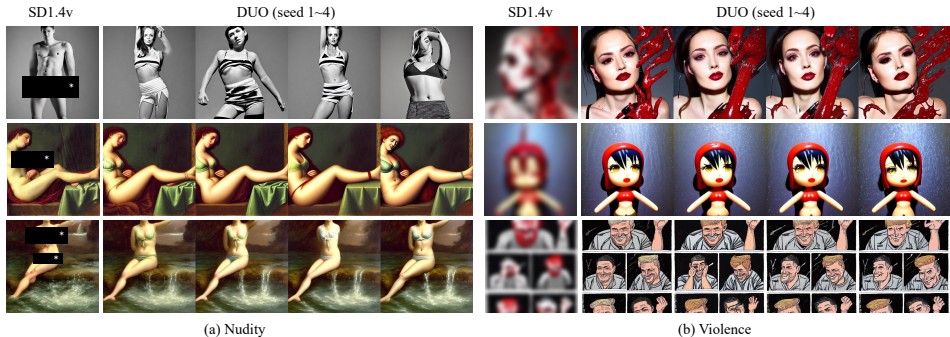

(a) Nudity                     (b) Violence

Figure 14: **Qualitative result on various seed numbers.** We used $\beta = 500$ for both nudity and violence. We use ▬ and blurring for publication purposes.

concept, which is closely related to "Blood". Table 3 below presents mean and standard devidation of LPIPS scores between 128 images generated from the unlearned model and the pretrained model. Lower scores indicate less impact on unrelated features. These results demonstrate that DUO effectively maintains the capability to generate visually similar but unrelated concepts compared to the existing methods. Figure 15 shows qualitative results of prior preservation for visually similar concepts.

**Stable-Diffusion 3** In Figures 16, and 17, we successfully applied DUO to Stable Diffusion 3 (SD3), which uses the transformer-based mmDiT architecture [10]. The evaluation process for SD3 was identical to that used for SD1.4v. In the Pareto curve, each point from left to right represents $\beta \in \{100, 250, 500, 1000, 2000\}$, with a consistent learning rate of 3e-5 across all experiments.

Table 3: **Impact of unlearning on visually similar concept generation.** We report the mean $\pm$ standard deviation of LPIPS scores between 128 images generated from the unlearned model and those from the pretrained model.

| Unlearned Concept | Safe Concept | ESD | UCE | SPM | DUO (black box) | DUO (white box) |
|---|---|---|---|---|---|---|
| Nudity | Woman | $0.58 \pm 0.11$ | $0.42 \pm 0.16$ | $0.31 \pm 0.15$ | $0.33 \pm 0.12$ | $0.55 \pm 0.17$ |
| | Man | $0.58 \pm 0.10$ | $0.31 \pm 0.17$ | $0.15 \pm 0.13$ | $0.14 \pm 0.09$ | $0.20 \pm 0.13$ |
| Violence | Ketchup | $0.69 \pm 0.15$ | $0.51 \pm 0.16$ | $0.20 \pm 0.15$ | $0.23 \pm 0.12$ | $0.35 \pm 0.14$ |
| | Tomato sauce | $0.58 \pm 0.19$ | $0.38 \pm 0.16$ | $0.11 \pm 0.13$ | $0.18 \pm 0.12$ | $0.28 \pm 0.12$ |
| | Strawberry jam | $0.56 \pm 0.13$ | $0.42 \pm 0.15$ | $0.13 \pm 0.12$ | $0.20 \pm 0.11$ | $0.31 \pm 0.12$ |

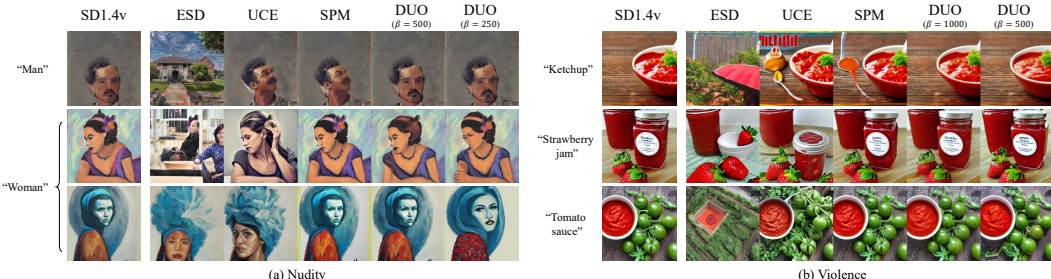

(a) Nudity        (b) Violence

Figure 15: **Quantitative result on generating close concept related to removed concept.** (a) Results of generating "man" and "woman" after removing nudity. (b) Results of generating "ketchup", "strawberry jam" and "tomato sauce" after removing violence.

Table 4: FID for SD3 unlearned nudity.

| Method | FID ($\downarrow$) |
|---|---|
| SD3 | 21.83 |
| DUO ($\beta = 500$) | 21.26 |
| DUO ($\beta = 250$) | 20.49 |

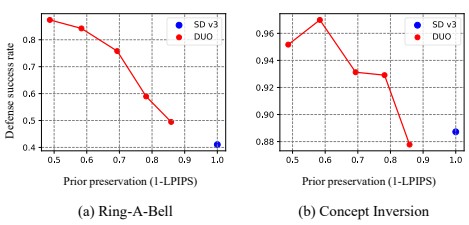

(a) Ring-A-Bell      (b) Concept Inversion

Figure 16: **Quantitative result of SD3 on nudity.** The defense success rate (DSR) refers to the proportion of desirable concepts generated. Prior preservation represents the perceptual similarity of images generated by the prior model and the unlearned model.

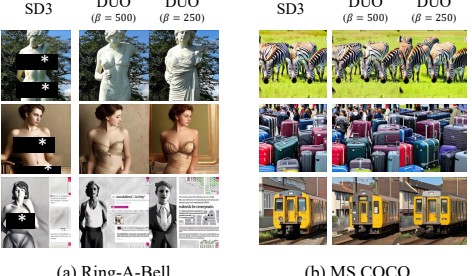

(a) Ring-A-Bell      (b) MS COCO

Figure 17: **Qualitative result of SD3 on nudity.** DUO effectively removes nudity while preserving the model's ability to generate unrelated concepts. We use ▬ for publication purpose.

Additionally, Table 4 shows FID scores on MS COCO. It demonstrates that applying DUO to SD3 maintains comparable fidelity to the pretrained model.

## C    Implementation details

### C.1    Dataset generation

**Nudity**    For generating the unsafe images $x^-$, we use "naked" as prompt with a guidance strength of 7.5. For $x^+$, we use "dressed" as with a strength of 7.5, and use "naked" as a negative prompt. We generate 256 images. We use nudenet [2] to regenerate images if there was no nudity in $x^-$ or if there was nudity in $x^+$.

**Violence**    To generate violence images we use four sub-concepts: Blood, Suffering, Gun, and Horror. To generate unsafe images $x^-$, we use the prompt: "a man, bleeding", "a man, suffering", "a man, with a gun", "horror". We generate 64 image per each prompt, total 256 images. For safe images $x^+$, we use the prompt: "a man, in comfort", "a man, enjoying", "a man, with peace", "comfort", with using unsafe prompt as a negative prompt.

**SDEdit**   When we use SDEdit, the magnitude of the added noise is $t = 0.75T$, where T is the maximum diffusion timesteps, and the guidance scale used is 7.5.

## C.2   Optimization.

**DUO**   Since $\beta$ linearly multiplies the learning rate at the time near initialization, we rescale the learning rate, dividing it by the same amount each time we increase $\beta$. Using $\beta = 100$ as a baseline, we use a learning rate of $3 \times 10^{-}4$, the batch size of 4, and LoRA rank of 32. When conducting violence unlearning, we generally observe that the training is more unstable compared to nudity, and we mitigate this by training LoRA to unlearn each concept and combine them as violence LoRA. Exploring the relationship between dataset characteristics and optimization stability in DUO is an interesting avenue for future work.

**Concept Inversion**   For Textual Inversion training, we used the Adam optimizer with a learning rate of $5 \times 10^{-}3$, batch size 4, and 3000 gradient steps. We used the same hyperparameters for all unlearning models.

# D   Derivation

## D.1   Deriving the Optimum of the KL-Constrained Reward Maximization Objective

For completeness, we provide the derivation of the solution of KL-constrained reward maximization here. For more detailed motivation and explanation of the derivation process, we refer readers to the DPO paper [44] Appendix A.1.

**Eq. (4) $\rightarrow$ Eq. (5)**   Start from Eq. (4).

$$\max_{p_\theta} \mathbb{E}_{x_0 \sim p_\theta(x_0)}[r(x_0)] - \beta \mathbb{D}_{\text{KL}}[p_\theta(x_0)||p_\phi(x_0)]$$

$$= \max_{p_\theta} \mathbb{E}_{x_0 \sim p_\theta(x_0)}\left[r(x_0) - \beta \log \frac{p_\theta(x_0)}{p_\phi(x_0)}\right]$$

$$= \min_{p_\theta} \mathbb{E}_{x_0 \sim p_\theta(x_0)}\left[\log \frac{p_\theta(x_0)}{p_\phi(x_0)} - \frac{1}{\beta}r(x_0)\right]$$

$$= \min_{p_\theta} \mathbb{E}_{x_0 \sim p_\theta(x_0)}\left[\log \frac{p_\theta(x_0)}{\frac{1}{Z}p_\phi(x_0) \exp\left(\frac{1}{\beta}r(x_0)\right)} - \log Z\right]$$

where we have partition function:

$$Z \triangleq \sum_{x_0} p_\phi(x_0) \exp \frac{1}{\beta}r(x_0)$$

.

Now define the probability $p^*(x_0)$ as:

$$p^*(x_0) \triangleq \frac{1}{Z(x_0)}p_\phi(x_0) \exp\left(\frac{1}{\beta}r(x_0)\right)$$

which is a valid probability distribution as $p^*(x_0) \geq 0$ for all $x_0$ and $\sum_{x_0} p(x_0) = 1$. Since $Z$ is not a function of $x$, we can then re-organize the final objective as:

$$\min_{p_\theta} \mathbb{E}_{x_0 \sim p_\theta(x_0)}\left[\log \frac{p_\theta(x_0)}{p^*(x_0)} - \log Z\right]$$

Now, since $Z$ does not depend on $p_\theta$, the minimum is achieved by $p_\theta(x_0) = p^*(x_0)$. Hench we have the optimal solution:

$$p_\theta(x_0) = p^*(x_0) = p_\phi(x_0) \exp\left(r(x_0)/\beta\right)/Z$$

## D.2 Diffusion-DPO

For completeness, we provide the derivation of Diffusion-DPO here. For more detailed motivation and explanation of the derivation process, we refer readers to the Diffusion-DPO paper [57].

**Eq. (10) → Eq. (11)**  Start from Eq. (10).

$$L_{\text{Diffusion-DPO}} = -\mathbb{E}_{x_0^+, x_0^-} [\log \sigma(\mathbb{E}_{p_\theta(x_{1:T}^+|x_0^+), p_\theta(x_{1:T}^-|x_0^-)}[\beta \log \frac{p_\theta^*(x_{0:T}^+)}{p_\phi(x_{0:T}^+)} - \beta \log \frac{p_\theta^*(x_{0:T}^-)}{p_\phi(x_{0:T}^-)}])] \quad (14)$$

Since $-\log \sigma(\cdot)$ is a convex function, we can utilize Jensen's inequality.

$$L_{\text{Diffusion-DPO}} \leq -\mathbb{E}_{x_0^+, x_0^-, p_\theta(x_{1:T}^+|x_0^+), p_\theta(x_{1:T}^-|x_0^-)}$$
$$\left[\log \sigma([\beta \log \frac{p_\theta^*(x_{0:T}^+)}{p_\phi(x_{0:T}^+)} - \beta \log \frac{p_\theta^*(x_{0:T}^-)}{p_\phi(x_{0:T}^-)}])\right] \quad (15a)$$
$$\approx -\mathbb{E}_{x_0^+, x_0^-, q(x_{1:T}^+|x_0^+), q(x_{1:T}^-|x_0^-)}$$
$$\left[\log \sigma([\beta \log \frac{p_\theta^*(x_{0:T}^+)}{p_\phi(x_{0:T}^+)} - \beta \log \frac{p_\theta^*(x_{0:T}^-)}{p_\phi(x_{0:T}^-)}])\right] \quad (15b)$$

In Eq. (15b), the intractable $p_\theta(x_{1:T}^+|x_0^+)$ and $p_\theta(x_{1:T}^-|x_0^-)$ are approximated by the diffusion forward process $q(x_{1:T}|x_0)$. Now, let us apply the product rule, i.e., $p(x_{0:T}) = p(x_T) \prod_{t=1}^T p(x_{t-1} \mid x_t)$, to $p_\theta(x_{0:T})$ and $p_\phi(x_{0:T})$.

$$L_{\text{Diffusion-DPO}} \leq -\mathbb{E}_{x_0^+, x_0^-, q(x_{1:T}^+|x_0^+), q(x_{1:T}^-|x_0^-)}$$
$$\left[\log \sigma([\beta \sum_{t=1}^T \log \frac{p_\theta^*(x_{t-1}^+|x_t)}{p_\phi(x_{t-1}^+|x_t)} - \log \frac{p_\theta^*(x_{t-1}^-|x_t)}{p_\phi(x_{t-1}^-|x_t)}])\right]$$
$$= -\mathbb{E}_{x_0^+, x_0^-, q(x_{1:T}^+|x_0^+), q(x_{1:T}^-|x_0^-)}$$
$$\left[\log \sigma([\beta \sum_{t=1}^T \left( \begin{array}{c} \log p_\theta^*(x_{t-1}^+|x_t)/q(x_{t-1}^+|x_t) - \log p_\phi^*(x_{t-1}^+|x_t)/q(x_{t-1}^+|x_t) \\ -\log p_\theta^*(x_{t-1}^-|x_t)/q(x_{t-1}^-|x_t) + \log p_\phi^*(x_{t-1}^-|x_t)/q(x_{t-1}^-|x_t) \end{array} \right)])\right]$$
$$= -\mathbb{E}_{x_0^+, x_0^-, q(x_t^+|x_0^+), q(x_t^-|x_0^-)}$$
$$\left[\log \sigma([\beta \sum_{t=1}^T \left( \begin{array}{c} \mathbb{D}_{\text{KL}}(q(x_{t-1}^+|x_t)\|p_\theta^*(x_{t-1}^+|x_t)) - \mathbb{D}_{\text{KL}}(q(x_{t-1}^+|x_t)\|p_\phi^*(x_{t-1}^+|x_t)) \\ -\mathbb{D}_{\text{KL}}(q(x_{t-1}^-|x_t)\|p_\theta^*(x_{t-1}^-|x_t)) + \mathbb{D}_{\text{KL}}(q(x_{t-1}^-|x_t)\|p_\phi^*(x_{t-1}^-|x_t)) \end{array} \right)])\right]$$

Finally, let us express the $\mathbb{D}_{\text{KL}}(q\|p)$ using the $L_{\text{dsm}}$, i.e., $\|\epsilon - \epsilon_\theta\|_2^2$. According to Eq. (48) from Kingma et al. [24], the following relationship holds:

$$\mathbb{E}_{x_t \sim q(x_t|x_0)} \left[ \mathbb{D}_{\text{KL}}(q(x_{t-1}|x_t)\|p_\theta^*(x_{t-1}|x_t)) \right] = \mathbb{E}_{\epsilon \sim N(0,I)} w(\lambda_t) \|\epsilon - \epsilon_\theta(x_t)\|_2^2 \quad (17)$$

where $\lambda_t$ is the signal-to-noise ratio (SNR), defined as $\lambda_t = \frac{\alpha}{1-\alpha}$, and $w$ is a coefficient factor that depends on the SNR, specifically $w(\lambda_t) = \frac{1}{\lambda_t} \frac{d}{dt} \lambda_t$. In practice, when training the diffusion model, $w$ is often treated as a constant [21]. By using Eq. (17) to replace each $\mathbb{D}_{\text{KL}}(q(x_{t-1}|x_t)\|p_\theta^*(x_{t-1}|x_t))$, we obtain the following:

$$L_{\text{Diffusion-DPO}} \leq - \mathbb{E}_{x_0^+, x_0^-, x_t^+ \sim q(x_t^+|x_0^+), x_0^- \sim q(x_t^-|x_0^-)}$$
$$\left[\log \sigma \left( -\beta T \omega(\lambda_t) \left( \begin{array}{c} \|\epsilon - \epsilon_\theta(x_t^+, t)\|_2^2 - \|\epsilon - \epsilon_\phi(x_t^+, t)\|_2^2 \\ - (\|\epsilon - \epsilon_\theta(x_t^-, t)\|_2^2 + \|\epsilon - \epsilon_\phi(x_t^-, t)\|_2^2) \end{array} \right) \right) \right] \quad (18)$$

