# OpenReview forum: "Direct Unlearning Optimization for Robust and Safe Text-to-Image Models"
_NeurIPS.cc/2024/Conference — NeurIPS 2024 poster_

### Official Review · Reviewer_huMh · 2024-07-05

**Soundness:** 3
**Presentation:** 3
**Contribution:** 2
**Rating:** 6
**Confidence:** 3

**Summary:**

The paper proposes a new method for unlearning diffusion-based generative models. The proposed method uses the basic idea of reinforcement learning with human feedback. To collect pairs of images to be unlearned and corrected for preference optimization, the author uses SDEdit to modify the NSFW content. Experimental results show that the proposed model successfully unlearns harmful content from the generated images.

**Strengths:**

The use of SDEdit for curating semantically similar paired images sounds novel and effective.

**Weaknesses:**

- Although this paper shows some promising results about diffusion model unlearning, the main technical contribution is limited to a combination of SDEdit and DiffDPO.
- It is clearly an overstatement that the preference optimization does not affect the quality of the unrelated concepts (in line 149).
- I have concerns about the reproducibility of this work.
- Minor
    - The main text and appendix have multiple typos and require careful proofreading.
    - Typo: Eq 17 → Eq 11 in the main text
    - Type: line 275: lambda?
    - Is DCO in line 285 a typo?

**Questions:**

- The reason why the gradient ascent term reduces the model's denoising ability, even in the presence of the KL term, is unclear. Could you provide more details on how adding a L_prior loss, which minimizes the difference between denoising directions at time T, can address this issue? The explanation is insufficient to grasp the logic behind the loss.
- What does ‘prior’ mean in ‘prior preservation performance’? If LPIPS measures the perceptual similarity between images, can we just use perceptual similarity? Since the prior in ‘prior preservation performance’ and the prior in ‘L_prior’ are different, it’s a bit confusing to follow section 4.3.

**Limitations:**

Limitations are well-addressed.

---

> ### Author Rebuttal · Authors · 2024-08-06
>
> Thank you for your detailed review and for recognizing the novelty and effectiveness of our approach in using SDEdit for curating paired images.
>
> ---
>
> > *[W1] Novelty of Technical Contribution*
>
> We want to clarify that our main contribution is proposing **a new image-based unlearning framework** to overcome the limitations of the previous prompt-based approach, not just combining two well-known techniques, SDEdit and DPO.
>
> As our framework is new, we provide **a new perspective on unlearning by formulating the diffusion models unlearning problem as preference optimization**. This perspective naturally arose from our image-based approach and offers a fresh paradigm for tackling unlearning challenges.
>
> We then introduce the **a output-preserving regularization** to specifically preserve unrelated features, a crucial aspect not addressed in standard preference optimization. Additionally, we adapt preference optimization for unlearning by using SDEdit to generate paired data, eliminating the need for human labeling.
>
> In summary, our work introduces a novel image-based unlearning framework, reframes unlearning as preference optimization, and incorporates output-preserving regularization. These innovations collectively offer a robust, efficient approach to diffusion model unlearning, addressing key limitations of existing methods.
>
> ---
>
> > *[W2] How can we ensure that DUO does not affect unrelated features?*
>
> We acknowledge that claiming DUO has absolutely no impact on unrelated features would be an overstatement. However, our comprehensive evaluations demonstrate that DUO significantly reduces the impact on unrelated features compared to existing baselines.
>
> As detailed in the global rebuttal (paragraph 3 and Fig. R3), we conducted comprehensive evaluations of DUO's prior preservation performance. Our results demonstrate that DUO maintains model prior for unrelated concepts while effectively removing unwanted content.
>
> In our revised manuscript, we will provide a more nuanced discussion, emphasizing that DUO significantly reduces impact on unrelated features compared to existing baselines, while still acknowledging the potential for minor, unintended effects.
>
> ---
>
> > *[W3] Reproducibility*
>
> We have taken several steps to address reproducibility concerns:
>
> - **Robustness to Random Seeds**
> We have conducted experiments using different random seeds (1 to 4) to demonstrate the stability of our results. Figure R5 in the global rebuttal PDF shows that DUO's Pareto curve is not sensitive to random seed variation and consistently outperforms baseline methods. Figure R6 provides qualitative evidence of similar unlearning results across different seeds.
>
> - **Code Release**
> To ensure full reproducibility and contribute to the field's advancement, we commit to releasing our unlearning and evaluation code upon publication.
>
> We believe these measures significantly enhance the reproducibility of our work. We welcome any suggestions for additional experiments or information that could further address reproducibility concerns.
>
> ---
>
> > *[W4] Minor errors*
>
> Thank you for bringing these typos and grammatical errors to our attention. We have made the following corrections:
>
> - L104: DSO → DUO
> - L175, 180, 506: Eq 17 → Eq 11
> - L275: lambda → output-preserving regularization
> - L285: DCO → DPO
> - L271: Ulike → Unlike
> - Figure 6: Quanlitative → Qualitative
>
> We appreciate your careful review. These corrections will be reflected in the final manuscript.
>
> ---
>
> > *[Q1] Logic Behind L_prior Loss*
>
> The L_prior loss, i.e., output preservation regularization, was designed to maintain the model's prior effectively, even when the KL divergence regularization is weak. While DPO's KL divergence regularization theoretically prevents significant deviation from the pretrained model's distribution, there's a trade-off:
>
> - Strong KL regularization hinders effective unlearning.
> - Weak KL regularization can cause the model to lose its denoising capability due to the gradient ascent term.
>
> L_prior addresses this by **enforcing output preservation for specific inputs**, particularly those unrelated to unsafe concepts.
>
> Consider $x_t^- = \sqrt{\alpha_t} x_0^- + \sqrt{1 - \alpha_t} \epsilon$, where $x_0^-$ represents an unsafe image and $\epsilon \sim N(0, I)$. Our goal is to unlearn features related to $x_0^-$ while maintaining the model's behavior for features related to $\epsilon$. L_prior achieves this by preserving outputs for $x_T = \epsilon$, effectively maintaining the model prior for unrelated concepts while allowing unlearning of unsafe content.
>
> Figure 8 demonstrates that L_prior's effect becomes more pronounced as beta decreases, highlighting its importance in maintaining model performance on safe concepts.
>
> ---
>
> > *[Q2] Meaning of Prior Preservation (1-LPIPS)*
>
> Prior preservation is a crucial objective in unlearning, aiming to maintain the model's performance on concepts unrelated to the unlearned content.
>
> To quantify this, we measure how much the model's output changes for unrelated concepts after unlearning. This is done by generating images using MS COCO prompts with both the pretrained and unlearned models. We use LPIPS to measure the perceptual difference between these generated images. Prior preservation is defined as 1-LPIPS. A higher value indicates that the unlearned model generates perceptually similar images to the pretrained model when given the same prompts and initial noise.
>
> This metric helps us ensure that our unlearning method effectively removes unwanted content without significantly impacting the model's performance on unrelated topics.
>
> To avoid confusion in the final manuscript, we will change the notation from L_prior to L_output for the output-preserving regularization term, distinguishing it clearly from the prior preservation metric.

---

> > ### Comment · Reviewer_huMh · 2024-08-09
> >
> > Thanks for the detailed responses with additional experiments. After reading the other reviews and rebuttal, I have decided to raise my score to 6. I hope a future revision will address the potential limitations and include additional experiments.

---

> > > ### Author Response · Authors · 2024-08-09
> > >
> > > We are pleased that we have been able to address the reviewer's concerns. We sincerely appreciate the time and effort the reviewer dedicated to reviewing our paper and providing valuable feedback. We will incorporate these insights, including the new experiments presented in this rebuttal, into our final manuscript. Thank you for helping us improve the quality of our work.

---

### Official Review · Reviewer_rmhF · 2024-07-12

**Soundness:** 4
**Presentation:** 3
**Contribution:** 4
**Rating:** 6
**Confidence:** 4

**Summary:**

This paper proposes a diffusion unlearning optimization framework to achieve NSFW visual content removal in T2I models. Specifically, the authors develop an image-based unlearning method that utilizes curated paired image data (unsafe images and their corresponding safe images generated by the SDEdit model) for preference optimization. Additionally, they introduce a regularization term to preserve the denoising capability of the diffusion model. This work claims to be the first to apply preference optimization to the unlearning problem. Experimental results demonstrate that their method effectively removes unsafe visual concepts without significant performance degradation on unrelated topics.

**Strengths:**

1. The model is well-explained, and the paper is well-written.
2. The novel perspective of applying preference optimization to the unlearning problem is interesting and promising.
3. The red teaming results are encouraging.

**Weaknesses:**

1. The authors only fine-tuned Stable Diffusion v1.4 for evaluation, which employs U-Net for denoising. However, other popular text-to-image (T2I) diffusion models, such as PixArt[1] and Stable Diffusion v3, which use Transformers for denoising, are not mentioned. It is somewhat limited in discussing the proposed method on only one T2I model. Can the proposed method be applied to the above models?

2. The unsafe concept may introduce some unrelated visual features when generating images, such as image details. How can the authors ensure that the proposed method does not affect these unrelated features when removing unsafe content?

[1] PixArt-$\alpha$: Fast Training of Diffusion Transformer for Photorealistic Text-to-Image Synthesis. ICLR 2024

**Questions:**

Please refer to the above weakness section.

**Limitations:**

Refer to previous sections.

---

> ### Author Rebuttal · Authors · 2024-08-06
>
> Thank you for acknowledging our paper's clarity and the novelty of connecting preference optimization to unlearning.
>
> ---
>
> > *[W1] Applicability of DUO to Diffusion Transformers*
>
> We appreciate your inquiry about DUO's applicability to other architectures. We have successfully extended our evaluation to include Transformer-based models. As shown in Figures R1 and R2 in the global rebuttal PDF, DUO effectively removes unwanted concepts in Stable Diffusion 3, which employs a Transformer architecture (mmDiT).
>
> We will incorporate these findings into our revised manuscript to provide a more comprehensive evaluation of DUO's capabilities across different model architectures.
>
> ---
>
> > *[W2] Impact of DUO on Unrelated Features*
>
> We acknowledge that claiming DUO has absolutely no impact on unrelated features would be an overstatement. However, our comprehensive evaluations demonstrate that DUO significantly reduces the impact on unrelated features compared to existing baselines.
>
> As detailed in the global rebuttal (paragraph 3 and Fig. R3), we conducted comprehensive evaluations of DUO's prior preservation performance. Our results demonstrate that DUO maintains model prior for unrelated concepts while effectively removing unwanted content.
>
> In our revised manuscript, we will provide a more nuanced discussion, emphasizing that DUO significantly reduces impact on unrelated features compared to existing baselines, while still acknowledging the potential for minor, unintended effects.

---

> > ### Comment · Reviewer_rmhF · 2024-08-08
> > **Reply to Authors**
> >
> > Thanks for your reply. The authors have addressed my concerns.  After considering the comments from other reviewers and your reply, I plan to keep my rating.

---

> > > ### Author Response · Authors · 2024-08-08
> > >
> > > We are glad to see that the reviewer's concerns have been addressed. We appreciate the time and effort the reviewer dedicated to reviewing our paper and offering valuable feedback.

---

### Official Review · Reviewer_tvVC · 2024-07-14

**Soundness:** 2
**Presentation:** 3
**Contribution:** 2
**Rating:** 6
**Confidence:** 5

**Summary:**

The authors introduce synthesized image data and preference optimization for concept unlearning in diffusion models. Additionally, they consider the regularization of model preservation performance to ensure a balanced approach.

**Strengths:**

1. The presentation is clear and well-structured.
2. There is a solid theoretical basis for the proposed preference optimization.

**Weaknesses:**

1. The efficacy of the proposed method is heavily dependent on the quality and diversity of the synthesized image pairs. The utilization of a small dataset, consisting of only 64 pairs, raises significant concerns about the potential for overfitting. To enhance the credibility of the findings, it is imperative for the authors to conduct comprehensive ablation studies. These studies should explore the impact of different sets of synthesized image pairs on model performance, potentially revealing critical insights into the method's robustness and generalizability.

2. The current robustness assessments of the study are limited by the employment of relatively weak attack scenarios. For a more robust evaluation of the unlearned models, it is crucial to incorporate a stronger, white-box attack methodology. The use of UnlearnDiffAtk [1], a commonly applied tool for assessing the robustness of unlearned diffusion models, is recommended. This approach would not only adhere to contemporary research standards but also significantly bolster the validity of the results. Furthermore, detailed reporting of the attack success rate (ASR) associated with UnlearnDiffAtk would provide a more precise quantification of the models' resilience against sophisticated attacks.


[1] "To Generate or Not? Safety-Driven Unlearned Diffusion Models Are Still Easy to Generate Unsafe Images ... For Now", ECCV 2024.

**Questions:**

Check the weaknesses section for more details.

**Limitations:**

Effectiveness of the proposed method heavily relies on the synthesized image pairs and different synthesized image pairs might cause high performance variance.

---

> ### Author Rebuttal · Authors · 2024-08-06
>
> First, thank you for acknowledging that our paper is clear and well-structured.
>
> ---
>
> > *[W1] Impact of the number of synthesized image pairs on DUO performance*
>
> Thank you for your constructive comment. Figure R7 from the global rebuttal PDF demonstrates how varying the number of synthesized image pairs affects the Pareto curve. The figure clearly shows that when the number of pairs is less than 64, there is a noticeable improvement in the Defense Success Rate. However, increasing the number beyond 64 pairs does not yield significant changes in the Pareto curve.
>
> This analysis suggests that 64 pairs provide a good balance between performance and computational efficiency for our method. We will include this discussion in our final manuscript to address concerns about the dataset size and its impact on model performance.
>
> ---
>
> > *[W2] Stronger White-Box Red Teaming results*
>
> We appreciate your valuable suggestion to incorporate more robust evaluation techniques. We have conducted additional experiments using UnlearnDiffAtk [1], a state-of-the-art white-box attack method designed for assessing the robustness of unlearned diffusion models.
>
> As illustrated in Figure R4 of our global rebuttal PDF, **DUO achieves Pareto-optimal performance compared to existing baselines** when subjected to UnlearnDiffAtk. This means that DUO provides the best trade-off between maintaining model performance and resisting attacks, outperforming other methods in both aspects simultaneously.
>
> We will incorporate a detailed analysis of the UnlearnDiffAtk results in our final manuscript, including qualitative comparisons to baseline methods.
>
> [1]: Zhang et al., To Generate or Not? Safety-Driven Unlearned Diffusion Models Are Still Easy To Generate Unsafe Images ... For Now https://arxiv.org/abs/2310.11868

---

> > ### Comment · Reviewer_tvVC · 2024-08-08
> >
> > The authors' response has addressed my concerns, and I am raising my rating to 6. I recommend including the newly conducted experiments in the revision to enhance readers' understanding and provide additional insights.

---

> > > ### Author Response · Authors · 2024-08-09
> > >
> > > We are pleased that we have been able to address the reviewer's concerns. We sincerely appreciate the time and effort the reviewer dedicated to reviewing our paper and providing valuable feedback. We will incorporate these insights, including the new experiments presented in this rebuttal, into our final manuscript. Thank you for helping us improve the quality of our work.

---

### Official Review · Reviewer_Ub4z · 2024-07-15

**Soundness:** 3
**Presentation:** 4
**Contribution:** 3
**Rating:** 6
**Confidence:** 4

**Summary:**

The authors address the issue of adversarial attack to diffusion model generating inappropriate image contents in this paper, critiquing the previous work on unlearning technique unlearns harmful prompt but making themself vulnerable to adversarial prompt. Instead they propose the image-based unlearning technique to remove unsafe visual features while retain the performance to generate image for safe concepts. During the training, they propose the method to generate a pair of safe and unsafe content and employ preference optimization to encourage diffusion model to generate the safe counterpart. The present the result in the evaluation to show that their model is better at defending adversarial attacks such as SneakyPrompt, Ring-A-Bell and Concept Inversion.

**Strengths:**

- The paper is is structured in a clear and logical manner, making it easy to follow. Both the introduction and the related work sections are well-written providing comprehensive context for the problem and the clear overview of previous work.

- The necessity of addressing this problem is convincingly justified, and the proposed solution is closely aligned with this motivation. The derivation of the proposed method is elaborated in detail and is technically sound.

- The authors demonstrate strong results in the evaluation and show convincing results successfully defend adversarial attack. The present ablation study to prove the effectiveness of output preserving regularization.

**Weaknesses:**

- There is a missing detail in experiments setup. The authors mention in line 194 that they decompose the violence concept into four categories and apply DUO to each of them and then merge the final models. If I understand correctly, the authors apply DUO independently for the four concepts resulting in four final models with LoRA. Please clarify how merging works to combine the four models.

**Questions:**

Please clarify the missing detail

**Limitations:**

The authors discuss the limitation in the conclusion.

---

> ### Author Rebuttal · Authors · 2024-08-06
>
> First, thank you for your recognition of our paper's clear structure and strong evaluation results.
>
> ---
>
> > *[W1] Construction of Violence Unlearning LoRA*
>
> We appreciate the opportunity to clarify this process. As you correctly surmised, we applied DUO independently to four subcategories of violence: "blood", "suffering", "gun", and "horror". For each concept, we trained a separate LoRA using DUO.
>
> To create the final model, we merged these individual LoRAs. Specifically, if we denote the pretrained model weight as $W$, and the LoRA matrices trained on the $i$-th concept as $A_i$ and $B_i$, the merged weight matrix can be expressed as:
>
> $\tilde{W} = W+ \sum_i B_iA_i$
>
> This approach allows us to comprehensively address the violence concept while maintaining the efficiency of LoRA-based fine-tuning. We will provide a detailed explanation of this merging process in the appendix of our final manuscript to ensure full clarity and reproducibility.

---

### Author Rebuttal · Authors · 2024-08-06

We thank the reviewers for their insightful feedback, which we will integrate into the final manuscript.

We appreciate the acknowledgment of our paper's clear structure and presentation (Ub4z, tvVC, rmhF), the novelty of connecting preference optimization to unlearning (rmhF), the use of SDEdit for curating paired images (huMh), and our robust results in defending against adversarial attacks (Ub4z, rmhF).

---

### Applicability of DUO to Diffusion Transformers

**Reviewer *rmhF* asked DUO's applicability to diffusion models beyond U-Net architecture, e.g., diffusion transformers.**

We appreciate this important question. We have extended our evaluation to include Transformer-based models. As shown in Figures R1 and R2 of the global rebuttal PDF, **we successfully applied DUO to Stable Diffusion 3 (SD3)**, which uses the transformer-based mmDiT architecture. The evaluation process for SD3 was identical to that used for SD1.4v. In the Pareto curve, each point from left to right represents $\beta \in \{100, 250, 500, 1000, 2000\}$, with a consistent learning rate of 3e-5 across all experiments.

Additionally, FID scores on MS COCO demonstrate that applying DUO to SD3 maintains comparable fidelity to the pretrained model:

| SD3   | DUO ($\beta=500$)   | DUO ($\beta=250$) |
|:-----:|:-----:|:-----:|
| 21.83 | 21.26 | 20.49 |

These results confirm DUO's effectiveness across different diffusion model architectures, including Transformer-based models like SD3.

---

### Impact of DUO on Unrelated Features

**Reviewers *rmhF* and *huMh* asked how DUO affects unrelated features when removing unsafe content.**

We appreciate the opportunity to clarify this important aspect of our work. Our "prior preservation" metric (1-LPIPS) specifically measures the model's ability to maintain unrelated feature generation. This is evaluated by comparing images generated from MS COCO prompts using both pretrained and unlearned models. A high prior preservation score indicates that the unlearned model produces perceptually similar images to the pretrained model when given the same prompts and noise.

To further validate our approach, we evaluated the model's ability to generate visually similar but safe concepts. For example, we compare the generation of red images (e.g., ketchup, strawberry jam) after removing the Violence concept, which is closely related to "Blood". The table below presents mean and standard devidation of LPIPS scores between 128 images generated from the unlearned model and the pretrained model. Lower scores indicate less impact on unrelated features. These results demonstrate that DUO effectively maintains the capability to generate visually similar but unrelated concepts compared to the existing methods.

| unlearned concept | safe concept | ESD    | UCE    | SPM    | DUO (black box) | DUO (white box) |
|:-----------------:|:------------:|:------:|:------:|:------:|:---------------:|:---------------:|
| **Nudity**        | Woman        | 0.58 ± 0.11 | 0.42 ± 0.16 | **0.31 ± 0.15** | *0.33 ± 0.12*     | 0.55 ± 0.17     |
|                   | Man          | 0.58 ± 0.10 | 0.31 ± 0.17 | *0.15 ± 0.13* | **0.14 ± 0.09**     | 0.20 ± 0.13     |
| **Violence**      | Ketchup      | 0.69 ± 0.15 | 0.51 ± 0.16 | **0.20 ± 0.15** | *0.23 ± 0.12*     | 0.35 ± 0.14     |
|                   | Tomato sauce | 0.58 ± 0.19 | 0.38 ± 0.16 | **0.11 ± 0.13** | *0.18 ± 0.12*     | 0.28 ± 0.12     |
|                   | Strawberry jam | 0.56 ± 0.13 | 0.42 ± 0.15 | **0.13 ± 0.12** | *0.20 ± 0.11*     | 0.31 ± 0.12     |

**bold**: first place
*itelic*: second place

Additionally, Figure R3 in the PDF provides qualitative evidence of DUO's capability to maintain the capability to generate unrelated features.

However, we acknowledge that claiming DUO has absolutely no impact on unrelated features would be an overstatement. In our final manuscript, we will provide a more nuanced discussion, emphasizing that DUO significantly reduces impact on unrelated features compared to existing baselines, while still acknowledging the potential for minor, unintended effects.

---

### Comment · Area_Chair_fCvd · 2024-08-09
**Read the rebuttal and discuss with the authors**

Hi Reviewers,

The authors have submitted their rebuttal responses addressing the concerns and feedback you provided. We kindly request that you review their responses and assess whether the issues you raised have been satisfactorily addressed.

If there are any areas where you believe additional clarification is needed, please do not hesitate to engage in further discussion with the authors. Your insights are invaluable to ensuring a thorough and constructive review process.

Best
AC

---

### Decision · Program_Chairs · 2024-09-25

**Decision:**

Accept (poster)

**Comment:**

This paper was evaluated by four experts who generally praised its novel approach and robust experimental results. Although the initial review scores were mixed and several concerns were raised, the authors effectively addressed these issues during the rebuttal phase. Consequently, reviewers who had initially provided lower scores revised their assessments in favor of acceptance. With unanimous support for acceptance, AC deems the paper a strong candidate for acceptance.